# Revisiting the long-term decreasing trend of atmospheric electric potential gradient measured at Nagycenk, Hungary, Central Europe

Attila Buzás[1,2], Veronika Barta[1], Tamás Horváth[3], and József Bór[1]

[1]Institute of Earth Physics and Space Science (ELKH EPSS), Sopron, Hungary
[2]Doctoral School of Earth Sciences, Faculty of Science, Eötvös Loránd University, Budapest, Hungary
[3]Institute of Natural Resources and Forest Management, Faculty of Forestry, University of Sopron, Sopron, Hungary

**Correspondence:** Attila Buzás (atteus645@gmail.com)

**Abstract.** In 2003, a decreasing trend has been reported in the long-term (1962–2001) fair weather atmospheric electric potential gradient (PG) measured in the Széchenyi István Geophysical Observatory (NCK, $47°38'$ N, $16°43'$ E), Hungary, Central Europe. The origin of this reduction has been the subject of a long-standing debate, due to a group of trees near the measurement site which reached significant height since the measurements have started. Those trees have contributed to the lowering of the ambient vertical electric field due to their electrostatic shielding effect. In the present study, we attempt to reconstruct the true long-term variation of the vertical atmospheric electric field at NCK. The time-dependent shielding effect of trees at the measurement site was calculated to remove the corresponding bias from the recorded time series. A numerical model based on electrostatic theory was set up to take into account the electrostatic shielding of the local environment. The validity of the model was verified by on-site measurement campaigns. The changing height of the trees between 1962 and 2017 was derived from national average age-height diagrams for each year. Modelling the time-dependent electrical shielding effect of the trees at NCK revealed that local effects played a pivotal role in the long-term decrease. The results suggest that earlier attempts could not quantify the shielding effect of the trees at NCK accurately. In this work it is found that the reconstructed PG time series at NCK exhibits an increase between 1962 and 1997 followed by a decaying trend since 1997. It is pointed out that long-term variation in summertime and wintertime PG averages should be analyzed separately as these may contribute to trends in the annual mean values rather differently.

## 1 Introduction

The atmospheric electric potential gradient (PG) is the reverse of the vertical atmospheric electric field usually measured near the ground (i.e., at a height of 1–3 m) (Harrison, 2004, 2012). It is a physical quantity that has been recorded since the very beginning of atmospheric electricity research (Chalmers, 1967, pp. 3–11). The PG is probably the most easily measurable fundamental parameter of the atmospheric Global Electric Circuit (GEC). The GEC is a global framework of electric currents flowing between the ground and the lower ionosphere. The GEC is powered by the global thunderstorm activity (Rycroft et al., 2000). The PG has been widely utilized in various studies to investigate environmental phenomena that have connection to electrical processes. Such phenomena are related to, e.g., lithosphere–atmosphere coupling related to earthquakes (Silva et al., 2011), influences of extraterrestrial effects on the neutral atmosphere (Märcz, 1997), electricity-related effects of the ENSO (El

Niño Southern Oscillation) phenomenon (Märcz and Sátori, 2005), the effect of radioactive fallout on atmospheric electricity (Pierce, 1972; Dragović et al., 2020), etc. However, the PG is highly dependent on site-specific effects, too. Some of these effects either affect the local conductivity of the ambient air (e.g., aerosols and local ionization sources) (Lucas et al., 2017; Nicoll et al., 2019) or, like nearby conducting objects, shield and bias the local natural electric field that would be measured in ideal conditions (Benndorf, 1900; Lees, 1915; Arnold et al., 1965; Wan et al., 2013).

In order to use the PG as a reliable diagnostic tool for long-term changes in Earth's electrical environment, long-term continuous measurements are required at a site where local effects are negligible or can be properly corrected for. Just a few long-term PG datasets, recorded at sites which fulfil these requirements, have been published in the literature yet (Kubicki, 2000; Harrison, 2003; Kubicki et al., 2021). In this study, the PG dataset recorded in the Széchenyi István Geophysical Observatory near Nagycenk, Hungary, Central Europe (NCK, $47°38'$ N, $16°43'$ E) is considered. At NCK, atmospheric electricity measurements

commenced in 1962 and have been running quasi-continuously to date (Bencze and Märcz, 1981; Sátori et al., 2013; Bór et al., 2020).

A long-term decline in the fair weather PG time series obtained at NCK has been reported (Märcz and Harrison, 2003). Märcz and Harrison associated this reduction with a global decrease of the incoming Galactic Cosmic Rays (GCR). This explanation has been questioned by Williams et al. (2005) who pointed out the importance of taking into consideration the

40 time-dependent electrical shielding effect of trees in the vicinity of the measurement site. Based on electrostatic theory and numerical modelling, Williams et al. (2005) have concluded that the time-dependent screening effect of trees alone accounts for the decline. Although the presence of the shielding effect of trees has been acknowledged by Märcz and Harrison (2003), they have not attributed the whole reduction to this effect alone stating that "*local effects at Nagycenk are unlikely to have dominated the changes there*". They conducted on-site parallel PG measurements with two instruments and interpreted the corresponding

results so that the long-term decline can not be ascribed to the shielding effect alone (Märcz and Harrison, 2006). Märcz and Harrison questioned the applicability of the model used by Williams et al. (2005) stating that the geometry of their model does not represent the configuration of the measurement site properly. It was pointed out that the separation between the trees and the instrument and the height of the trees are different in the model and in the reality (Märcz and Harrison, 2006). Therefore, the question of the long-term decrease in the PG time series observed at NCK has remained open.

The principal aim of this study is to obtain the unbiased long-term time series of PG records at NCK by removing the systematic distortion caused by the time-dependent electrical shielding effect of trees growing near the measuring instrument. This is achieved by quantifying the shielding effect via setting up an electrostatic model of the environment of the measurement site using accurate local geometry and reconstructing the changes in the environment over the decades of the measurements as realistically as possible. Long-term variation of the PG in the obtained corrected time series at NCK is rather different from that

of the uncorrected one. This requires re-consideration of the weights of various environmental processes in the interpretation of long-term PG variations observed at NCK.

## 2 Research methodology

### 2.1 Details of the observatory and the data

The Széchenyi István Geophysical Observatory near Nagycenk, Hungary (NCK) was founded in 1956–1957 during the International Geophysical Year. The primary mission of the observatory is to monitor the Earth's electromagnetic environment continuously. Earth current and atmospheric electricity measurements (e.g., potential gradient and Schumann resonances), as well as ionospheric observations have been running there simultaneously for decades now (Sátori et al., 2013; Bór et al., 2020).

In NCK observatory, the PG has been monitored by a locally developed radioactive apparatus since 1962 (Bencze and Märcz, 1981). This instrument (*PG62*) is a so-called potential equalizer that contains a radioactive probe which ionizes the ambient air, thus increasing its conductivity (Chalmers, 1967, pp. 122–124 and pp. 128–134). The radioactive material is placed at a height of 1 m. The potential difference is measured between a grounded electrode and another electrode at ground level. Because of the enhanced conductivity of the ambient air, the potential acquired by the electrode at the ground is the same as that at 1 m height. The potential difference measured this way equals to the potential gradient within 1 m above the Earth's surface. Later on, in 1998, another radioactive instrument (*PG98*) has been installed with the purpose of studying the effect of nearby trees on the PG measurement (Märcz et al., 2001; Märcz and Harrison, 2006). Determination of the zero signal offset every day and weekly full instrument calibration in the $\pm 250\ V\ m^{-1}$ range have ensured high quality measurements over the course of the years of continuous operation. Detailed characteristics of the instruments and the applied calibration techniques were reported by Märcz et al. (2001). In 2013, a more state-of-the-art instrument, a Boltek EFM-100 electric field mill has been deployed in the observatory.

In the present study, the long-term PG time series obtained by the *PG62* apparatus is investigated exclusively. We consider only the so called fair weather PG data that are not affected by local thunderstorms and shower clouds. In order to select the fair weather data we were confined to use an approach based on the statistical distribution of the PG data at NCK as we lack the appropriate supplementary meteorological data (Harrison and Nicoll, 2018). Firstly, only PG values with positive polarity are retained as negative PG is characteristic to foul weather (Harrison and Nicoll, 2018). Secondly, an upper boundary for the fair weather PG value is determined in each year by calculating the median absolute deviation of the PG data in the given year and multiplying it by five as described in Lucas et al. (2017). This way, the selection procedure takes into account the year-to-year variation of the PG. Data that are uncertain for any reason or hourly averages that were taken just from a part of the hour were not included in this study. Then, annual, summer (June, July, August), and winter (January, February, December) means, as well as annual minima (the lowest of the monthly means in a given year) and maxima (the highest of the monthly means in a given year), were derived from these hourly averages. As an extension to the original dataset (1962–2001), records from years 2002–2009 were also included in this study. After 2009, the electronics of the *PG62* and *PG98* instruments were modernized sequentially which introduced gaps in the recorded data series. For consistency with the paper by Märcz and Harrison (2003), records after 2009 were not included in the present study. Even so, an extended time period from 1962 to 2009 was analyzed.

## 2.2 Model setup

A numerical model based on electrostatic theory has been set up to calculate the shielding effect of the trees. The concept of this model is similar to that which has been used by Williams et al. (2005). However, the authors of that paper applied a model geometry in which some important characteristics of the measurement site have not been taken into account. At NCK, the measurement site is located in a national park in a clearing embedded in a forest. The area is enclosed by trees along a two meter tall wire fence on the western side, by forest on the southern and western sides, a line of pine trees on the northern side,

and by another group of trees on the eastern side. Additionally, there is a building of 3.5 m height in the southern half of the clearing (Fig. 1a and b). In the model by Williams et al. (2005), however, only the eastern group of trees was included.

Our model was built using the electrostatics module of the FEMM 4.2 (Finite Element Method Magnetics) software package (Meeker, 2015). The program solves partial differential equations in two dimensions by the finite element method in points over a triangulated mesh. Cross sections of the site were modelled in both the east–west (Fig. 1c) and the north–south directions.

The trees at the western and southern sides of the field were represented by blocks extending practically to infinity away from the clearing (100 m in the model). The line of trees on the north and the eastern group of trees were modelled by blocks of finite width (10 m). The building was modelled by a block of 3.5 m height and of 6 m width in the north–south direction. In case of the eastern group of trees, there are some branches that have significant length. These branches were included in the model as well (Fig. 1c). The dimensions and locations of the objects in the model were those measured accurately in 2017.

Tree heights and branch lengths in the model were parameterized according to the growth of trees in years of the investigated time period (1962–2009, see in Sect. 2.4).

With these conditions, the geometry represents a potential well (Fig. 1c). The potential was set to 1200 V at the upper boundary at a height of 30 m. Without any conductive objects in the area, this corresponds to a uniform potential gradient of 40 V m$^{-1}$, which is a typical PG value measured at NCK in fair weather conditions. The lower boundary of the evaluation

region was kept at ground potential (0 V). Note that the upper and lower boundaries were kept at a uniform potential in all years between 1962 and 2009 as we have no data on the possible year-to-year variation of the conductivity of the ground at NCK and the ionospheric potential above the site. The resolution of the mesh used in the calculations was set to 10 cm.

One of the key parameters in the model is the dielectric constant of the objects. The dielectric constant of living trees is highly dependent on their moisture content. As we lack data on the moisture content of trees we are confined to use an average

dielectric constant derived from the literature which was chosen to be 50 (Tomasanis, 1990; Salas et al., 1994). Initially, a dielectric constant of 50 was assigned to all the trees. In case of the building, an average dielectric constant value of 15 was taken from measurements of the dielectric constant of concrete which exhibits a more moderate dependency on moisture compared to living trees (Jamil et al., 2013).

In this model, the shielding factor of the PG at any point is calculated as the ratio of the modelled vertical electric field

at that location and the unshielded 40 V m$^{-1}$ value. The total shielding is calculated as the superposition (product) of the shielding factors from the east–west and north–south crossection models as well as that from the model of the building. Also the building was modelled as a 2D object, but it was not included in the north–south cross section model of the area. It was

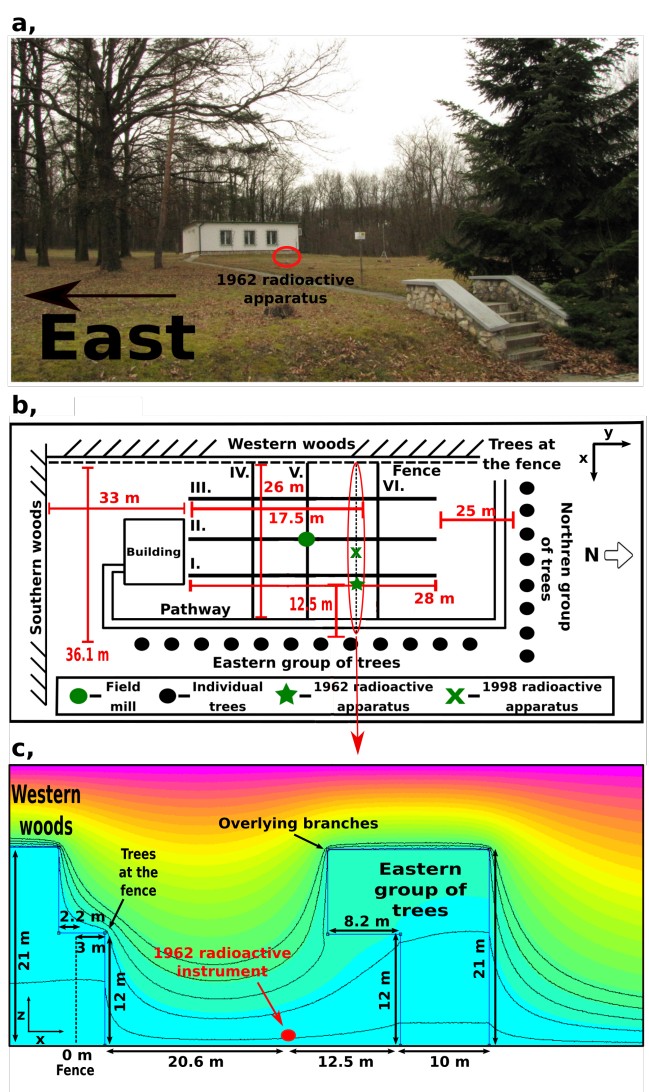

**Figure 1. (a)** Photo of the measurement site. **(b)** Schematic map of the measurement site (not to scale). "1962 radioactive apparatus" is the older radioactive instrument (*PG62*), "1998 radioactive apparatus" denotes the newer one (*PG98*). **(c)** Sketch of the model geometry in the east-west cross section, along lines IV, V, and VI (see panel *b*). Heights and distances correspond to the model for the year 2017. The distribution of electric potential as obtained from the model with effective tree conductivities (see Section 3.1) is indicated by colors. Light blue to magenta colors correspond to near zero to stronger electric potentials, respectively. Equipotential lines are drawn at 12, 50, 90, 125, and 170 V.

considered separately in order to take into account its finite extension (11.3 m) in the east–west direction. For the validation of the model (see Sect. 2.3), shielding factors were calculated in points along lines I-VI (Fig. 1b). Those shielding factors were calculated as a function of distance from the northern wall of the building (i.e., $y$ coordinates) in case of points along

lines I and II (Fig. 1b). For the points along line III and for all other points to west from line III, however, the trigonometric distance ($\sqrt{(x^2 + y^2)}$) from the north–western corner of the building was used. The $y$ distance between the northern wall of the building and the first points closest to the building along lines I-II was about 0.5 m. The trigonometric distance between the north–western corner of the building and the first point of line III to the south was about 4.9 m.

## 2.3  Validation of the model

In order to evaluate the performance of the model, its output was compared to test measurements made in NCK. Relative variation of the electric field was surveyed by two Boltek EFM-100 field mills. One of them was kept at a fixed location in the middle of the field, as far from any disturbing objects as much as it was possible (green dot in Fig. 1b), while the other was placed at different locations with 2 m spacing along lines I-VI in the clearing as indicated in Fig. 1b. The $x$ coordinates of lines I, II, and III are 22 m, 16 m, and 10 m, while the $y$ coordinates of lines IV, V, and VI are 6.5 m, 12.5 m, and 20.5 m, respectively. Both field mills were put at a height of 1.5 m and measured the PG simultaneously for about 3 minutes in each point. The relative variation of the electric field over the area is represented by ratios of the averaged PG measured by the fixed and the portable field mills. Due to the short measuring period, extra care was taken to count only those points in the average which exhibited similar variations in both time series.

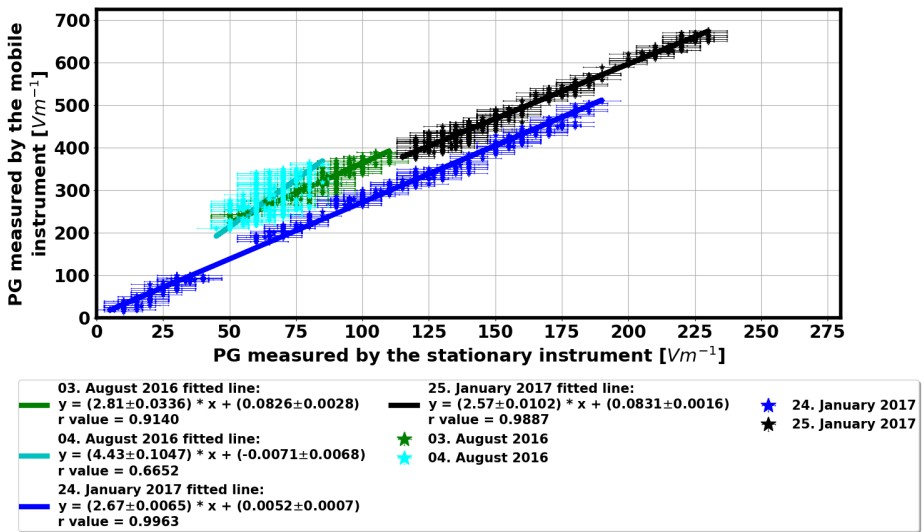

**Figure 2.** Intercalibration of the two field mills. Horizontal and vertical error bars represent the uncertainty of the field mills (7.1 $Vm^{-1}$ after resampling the data to 1 Hz).

Two test measurement campaigns were performed; one in summer (03–04 August 2016) and another one during the next winter (24–25 January 2017) with all points measured on each day so that altogether four datasets were obtained. There were fair weather conditions on all four days. The two field mills (the stationary and the mobile instrument) were intercalibrated

on each day of the campaign to ensure that no instrumental effects bias the results. For intercalibration, PG values measured simultaneously by the two instruments were compared, when the moved field mill was in neighbouring points of the fixed

instrument, and in an additional point towards north along line II (Fig. 1b). It was assumed that the field mills measure practically the same electric field in these points and their mounting poles do not bias the ambient electric field at the location of the other instrument. PG values measured simultaneously by the two field mills are shown in (Fig. 2). Note that the huge differences in magnitude between PG values measured by the two field mills can be attributed to the different sensitivity and instrumental drift of the instruments. Another factor that may have contributed to this difference is that the orientation of the

head of the field mills differed from each other. In case of the stationary field mill, the head was oriented downwards whereas in case of the mobile one, the head was pointed upwards. Different measuring head orientations distort the ambient atmospheric electric field in case of the two instruments differently. The field mill with upward orientated measuring head (the mobile instrument) measures higher PG values as equipotential lines are somewhat denser at the top of the mounting pole. On the other hand, the downward-faced stationary field mill measures smaller PG values as its mounting pole shields the ambient electric

field. Please note that we are not interested in the absolute PG values in this test but rather in the relative PG, the ratio of the PG measured by the two instruments after cross-calibrating the two field mills. According to the results, linear conversion functions could be used to correct for the differences between the two instruments in terms of sensitivity (slope) and baseline offset (y axis crossing). Line fitting was done via a method of multivariate linear regression analysis, the so called Orthogonal Distance Regression (Boggs and Rogers, 1990). This way, the uncertainty in both coordinates (i.e., in the PG values measured

by the mobile and stationary instruments) could be taken into account. The uncertainty range of the raw PG data is 10 $Vm^{-1}$ ($\pm 5\ Vm^{-1}$), which is the maximum resolution of the digital data, sampled at 2 Hz, from the EFM-100. Averaging the records to one sample per second lowers the uncertainty range at each point to 7.1 $Vm^{-1}$. The range of PG values recorded for intercalibration on 04. August was rather narrow and the noise in the data was relatively high so that the reliability of the linear parameters deduced with multivariate linear regression was rather poor. This resulted in higher uncertainty in case of data mea-

sured on this day (r=0.66, slope of the fitted line=4.43±0.1047 and y axis intercept=-0.0071±0.0068, see Fig. 2) compared to results from the other days. It is not readily clear whether the outstandingly high sensitivity on this day is a result of different measuring setup or it is only due to the poorly determined linear fit on the narrow data values range. The measurements were carefully set up similarly on all days. Nevertheless, please see the related discussion in Section 3.1. Note the differences in the relative baseline offset parameters on the four days. These differences may arise from assembling and disassembling of the

instruments and their masts slightly differently day to day in spite of careful handling. Generally, the goodness of the fit (r) was better (higher) for wintertime measurements because larger range was covered by the recorded values (Fig. 2) compared to summertime surveys. In any case, values measured by the moved field mill were always transformed by the corresponding conversion function before the relative field variation was calculated.

On the other hand, tree heights measured in 2016–2017 were set in the analytical model and the field is calculated at the

location of the fixed field mill as well as at each location of the moved instrument. The ratios of the values were then calculated and were compared to those from the field measurements (see Sec. 3.1).

Digitized PG records are available from both the *PG62* and the *PG98* instrument in several years between 1999 and 2007. The ratio of annual PG averages from the two instruments was also compared to the ratio of the modelled PG values at the location of the instruments in those years to check the reliability of the model retrospectively (see Sec. 3.2).

## 2.4 Time-dependency of tree heights

In order to determine the magnitude of the shielding effect of trees in the past, one has to know how the height of the trees changed over the decades of PG measurements. In the model, tree heights were considered separately for the eastern group of trees, for the trees at the fence, for the western and southern woods, and for the trees on the north side of the clearing (Fig. 1a, b). Species of the trees at those locations (turkey oak, gean tree, common ash tree, and pine) have been identified and their heights were calculated retrospectively using national average age-height curves for each type (Sopp, 1970; Solymos, 1971). The age of trees was estimated using administrative records of the area, which is a national park, archive photos from the observatory, and personal reminiscences. The western and southern woods were planted according to cultivation plans. Trees to the east and north were planted as regular saplings shortly after the construction of the observatory was finished, while the trees at the fence have grown up from seeds. An uncertainty of $\pm 1$ m was considered for all estimated tree heights (Fig. 3). This uncertainty is a resultant of many factors. One of the sources of this uncertainty is that the age and height of the trees in 2016–2017 could be only determined ambiguously. Another source is that we lack detailed information about the soil, climate, and ecological environment of the measurement site which would be needed to determine the tree heights more accurately. As we lack these information, we are confined to determine the tree heights based on national averages. The uncertainty of 1 m is an expectable upper estimation of the resultant uncertainty when all these factors are taken into account.

Note that the initial height of the eastern group of trees is 2.25 m in the model of Williams et al. (2005). However, according to our estimation based on archive observatory photos and personal reminiscences, it was more probably 3.5 m.

Moreover, we had to estimate the varying length and height of the overlying branches in case of the eastern group of trees (Fig. 1a and c). This problem was challenging as there are no data on the growth rate of the branches. The length and the height of the branches are 8.2 m and 12 m, respectively, as we measured it later in 2019. According to archive photos from the 1960's and the measurements in 2019, the ratio of the branch length and the tree height (ca. 0.39), as well as the ratio of the height of the branches and the tree height (about 0.57), were practically the same in 1962 and in 2019. Therefore, these ratios were used to calculate the length and height of the branches throughout the investigated time period (1962–2009). This way, it was possible to include the time variation of the additional shielding effect due to the branches in the model calculations.

## 3 Results

### 3.1 Model validation by on-site measurements in 2016–17

The performance of the model is demonstrated in detail along measurement lines I (Fig. 4a) and VI (Fig. 4b) which run the closest to the location of the older radioactive probe (*PG62*, Fig. 1b). Note that the originally measured values of relative PG

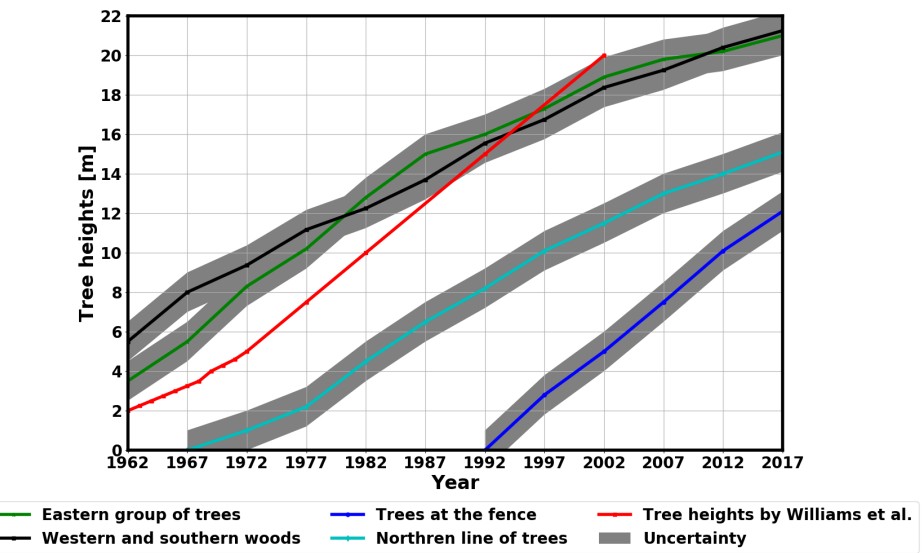

**Figure 3.** Time-dependent tree heights in NCK.

variations needed an additional correction before the plots in Fig. 4a and b showing the unbiased values were produced. The necessity of this additional conversion turned out during the evaluation of the measured PG values. Despite the measurements were done in fair weather conditions, originally recorded PG values at the beginning of lines I and II, i.e., very close to the northern wall of the building (Fig. 1b) were negative, even after applying the deduced conversion function. Model calculation confirmed that the electric field there cannot change its sign, so the only reason behind negative PG values can be that the field mill of fixed position must have had a negative baseline offset. Note that EFM-100 field mills are supposed to measure strong electric fields up to several $\mathrm{kV\,m^{-1}}$. A small baseline offset causes negligible relative error in high electric field conditions, but can result in considerable relative error for such damped, weak electric fields that were measured during our campaigns. Absolute calibration of the applied field mills were not available to us, so we handled this obvious bias so that an offset was added to all converted PG values as well as to those measured by the fixed instrument so that the smallest PG average right in front of the building became exactly zero. Relative PG values shown in Fig. 4a and b are ratios of these corrected (shifted) averages.

Relative PG values near the building along line I deduced from the measurements on 04 August, 2016 differ significantly from the corresponding values obtained on the other three days (Fig. 4a). Note that the conversion function for this day could be deduced only with a higher uncertainty than in case of the other three days of measurement (Fig. 2). The source of these anomalous values is likely the ambiguous conversion function determined for the data on 04 August. Nevertheless, values measured on 04 August were retained during the model evaluation because the deduced PG ratios show the same trend as those from the other three days and all values are fairly close to each other especially near the *PG62* instrument which is in the focus of our interest.

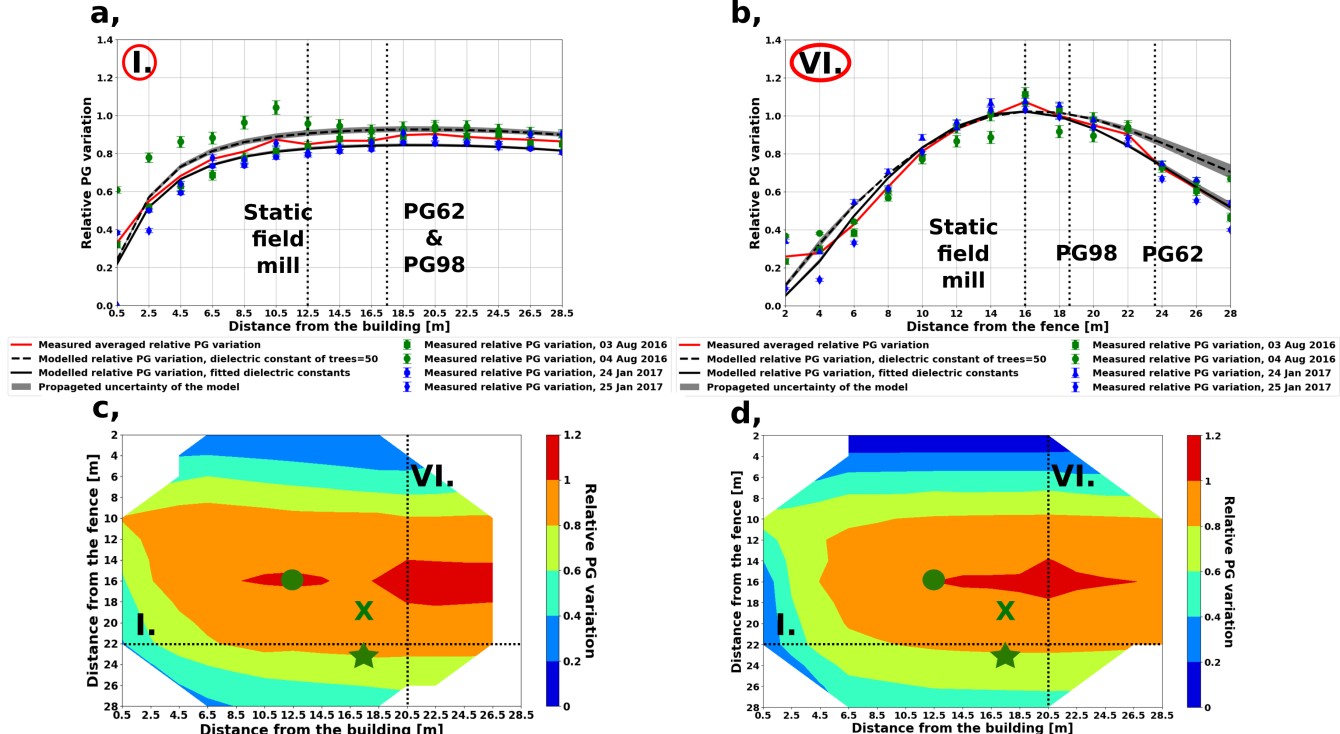

**Figure 4.** Measured and modelled relative PG variation along line I **(a)** and along line VI **(b)**. Measured **(c)** and modelled **(d)** relative PG variation interpolated over the grid of measurement points. Vertical dotted black lines on panels **(a)** and **(b)** denote the locations of the PG instruments. The green star, the green x, and the green dot on panels **(c)**, and **(d)** mark the position of the older radioactive apparatus (*PG62*), the newer radioactive apparatus (*PG98*), and the static field mill used as the reference point during the campaign measurements, respectively. See the site map in Fig. 1b for reference.

The mean of relative PG values obtained on 03–04 August, 2016 and 24-25 January, 2017 were calculated at each point of the measurement lines. The curve of measured relative PG variation was compared to the relative variation obtained from the model. Initially, the model was set up with carefully measured tree positions, heights, and branch length in 2017. The dielectric constants of the building and the trees were set to mean values derived from the literature, 15 in case of the building (Jamil et al., 2013) and 50 in case of the trees (Tomasanis, 1990; Salas et al., 1994) (Fig. 4a and b). It can be concluded that the model with mean dielectric constants derived from the literature underestimates the shielding effect most importantly near the *PG62* instrument (Fig. 4a and b).

To achieve better agreement between the model and the measurements, the dielectric constant of the trees and that of the building were adjusted. In the best fitting case, the effective dielectric constant for the trees/leaves is 130 at the eastern side and 120 at the western forest. These values are larger than the values 25-85 appearing in the literature for living trees and foliage (Tomasanis, 1990; Salas et al., 1994). The dielectric constant of living trees is highly dependent on the actual conditions of the trees, especially on their moisture content which was unknown to us. Note that the relatively simple geometry of the applied

model cannot reproduce the complex structure of the branches and leaves in the foliage. Taken into account the high uncertainty

of the actual physical conditions of the trees and the limitations in the complexity of our model, the best-fit effective dielectric constant were used in the model calculations. The effective (fitted) dielectric constant of the building was found to be 20, which is relatively close to published characteristic values (8-16) for concrete (Jamil et al., 2013). In case of the building, the higher dielectric constant can be explained by the fact that there are some metallic elements protruding from the roof of the building. Ultimately, with these effective dielectric constants, the modelled relative PG variation (solid black line) fits very well to the

measured PG relative variation (solid red line) both along line I and VI (Fig. 4a and b). At the location of the PG62 apparatus, the measured relative PG is larger than the modelled one only by about 0.12 %.

Figure 4c shows the relative PG variation interpolated over the whole surveyed area using values of the averaged campaign measurements obtained along lines I-VI (Fig. 1b). The result clearly indicates that the PG generally decreases towards the lines of trees both at the western and at the eastern side of the measurement site. The PG decreases also towards the building on the

southern side and a very small reduction also seems to appear as the points approach the northern line of pine trees (Fig. 4a). Such variations are in fact expected. When the field meter gets closer to the objects, the equipotential lines bend upward so that the vertical component of the electric field (i.e., the PG) decays quickly (Benndorf, 1900; Lees, 1915; Arnold et al., 1965; Wan et al., 2013).

These features are also present in the model (Fig. 4d) which reflects fundamental characteristics of the measured PG variation

(Fig. 4c) fairly well. There are some deviations between the measurements and the model perhaps most notably at the western side of the area. Taking into consideration that the field meter was put right into the dense bush there, these departures are understandable as the complex geometry of the dense scrub on the western side could not be taken into account completely (Fig. 4c and d).

## 3.2  Correction of the long-term PG measurements in NCK

The shielding effect of nearby trees on the PG measurements in NCK was corrected by the factors in Table 1. The shielding effect was eliminated from the data by multiplying the measured PG annual means by the correction factor in each year (Fig. 5a). The correction factor for a given year is the ratio of an unshielded, theoretical PG value derived from the model ($PG_0$ = 40 V m$^{-1}$) and the shielded PG value calculated from the model at the location of the older radioactive apparatus ($PG62$) at 1m height. The uncertainty of the corrected PG time series is inherited from the uncertainty of the tree height curves

(Fig. 3). Since the estimated height of the trees carries an uncertainty of $\pm 1$ m, results from the model were calculated with both maximum and minimum tree heights within the uncertainty range in each year in order to obtain the corresponding uncertainty range of the corrected PG values. The model was always set up with the appropriate tree heights in the given year ($PG_{Year}$). The $PG_0/PG_{Year}$ ratio was calculated with a temporal resolution of 5-years due to the fact that tree heights were determined in this time resolution. Correction factors were interpolated using cubic splines to have 1-year resolution. Since the model

was validated in 2016–17 with measurements done at 1.5 m height, the difference in the shielding in 1 m and 1.5 m heights was calculated in the model at the locations of the $PG62$ and the $PG98$ instruments in all considered years. The maximum

differences during the 1962-–2017 time period were found to be 0.59 % and 0.55 %, respectively, so no significant bias can be expected in the results because of this difference.

Correction factors used by Williams et al. (2005) are shown in Table 1 for reference. The difference between the two approaches is significant. The model in this study yields from $\sim 8\pm4.3$ % (1962) up to $\sim 43\pm3.3$ % (2002) larger shielding effect for the years 1962 to 2002 than the approach of Williams et al. (2005).

**Table 1.** Time-dependent correction factors and modelled PG reduction at NCK exclusively due to the shielding effect of trees. The percental reduction shows how much the PG diminished in each year compared to the measured annual mean in 1962 solely due to the shielding effect. The uncertainties are calculated from the uncertainties in the tree heights and carry the uncertainties of the model (Fig. 3).

| Year | Correction factor ($PG0/PG_{Year}$), present study | Percental PG reduction caused by the shielding effect, present study | Correction factor ($PG0/PG_{Year}$) by Williams et al. (2005) | Percental PG reduction caused by the shielding effect, Williams et al. (2005) |
|---|---|---|---|---|
| 1962 | 1.13±0.05 | 11±3.8 % | 1.04 | 3.85 % |
| 1967 | 1.25±0.07 | 20±4.5 % | 1.07 | 6.53 % |
| 1972 | 1.47±0.11 | 32±4.8 % | 1.12 | 10.63 % |
| 1977 | 1.69±0.13 | 41±4.6 % | 1.2 | 16.57 % |
| 1982 | 2.04±0.16 | 51±3.9 % | 1.3 | 22.65 % |
| 1987 | 2.38±0.17 | 58±3.1 % | 1.4 | 28.32 % |
| 1992 | 2.57±0.18 | 61±2.7 % | 1.5 | 33.54 % |
| 1997 | 2.81±0.17 | 64±2.4 % | 1.62 | 38.34 % |
| 2002 | 3.09±0.17 | 68±1.9 % | 1.75 | 42.73 % |
| 2007 | 3.26±0.16 | 69±1.6 % | - | - |
| 2009 | 3.31±0.16 | 72±1.5 % | - | - |

Uncertainty of the corrected PG time series is also inherited from the uncertainty of the tree height curves (Fig. 3). Since the estimated height of the trees carries an uncertainty of $\pm 1$ m, results from the model were calculated with both maximum and minimum tree heights within the uncertainty range in each year in order to obtain the corresponding uncertainty range of the corrected PG values.

By incorporating available quality checked measurements from the *PG98* instrument in the analysis, the result of the applied correction can be further validated retrospectively. We can calculate the ratio of the PG values measured by the newer and older radioactive apparatuses (*PG98/PG62*) in years when both are available, and compare these ratios to the modelled ones.

**Table 2.** Measured and modelled $PG_{98}/PG_{62}$ ratios. Maximum of the KDE denotes the location of the maximum of the Gaussian Kernel Density Estimation function. Where both the mean and the KDE-based estimation of the $PG_{98}/PG_{62}$ ratio were available, the one at the maximum of the KDE values were used to compare the measured ratios with the modelled ones. Measured ratios printed in bold were used in the comparison.

| Year | Mean of the measured $PG_{98}/PG_{62}$ ratios | Measured $PG_{98}/PG_{62}$ ratio at the maximum of the KDE | Standard deviation of the measured ratios | Modelled $PG_{98}/PG_{62}$ ratios | The ratio of the measured and modelled $PG_{98}/PG_{62}$ ratios |
|------|------|------|------|------|------|
| 1999 | 1.41 | **1.29** | 0.57 | 1.34 | 0.96 |
| 2000 | 1.34 | **1.3** | 0.28 | 1.34 | 0.97 |
| 2001 | 1.37 | **1.29** | 0.39 | 1.34 | 0.96 |
| 2002 | 1.36 | **1.25** | 0.34 | 1.34 | 0.93 |
| 2007 | 1.43 | **1.34** | 0.59 | 1.32 | 1.01 |
| 2017 | **1.28** | - | 0.24 | 1.29 | 0.99 |

The results of this investigation are presented in Table 2. The modelled and the measured $PG98/PG62$ ratios are in a good agreement, especially if the measured standard deviations are taken into account. The difference between the modelled and measured averages does not exceed 7 %. This result further assures that the model mirrors the real variations caused by the shielding effect in the past.

## 4 Discussion

### 4.1 On the long-term variation of the corrected PG time series in NCK

Time-dependent shielding effect modelled in this study was compared to that produced by Williams et al. (2005) (Fig. 5a, Table 1 and 3). The shielding effect modelled by Williams et al. is significantly smaller in each year than the one calculated in this study (Table 1). This difference can be attributed to the different model geometries used in the two studies. Williams et al. did not include the impact of the western and southern woods, the northern line of trees, the overlying branches at the eastern side of the field, and the building to the south which all had an effect on the reduction (Fig. 1b and c). Moreover, the distance between the 1962 radioactive instrument and the eastern group of trees was 20 m in the calculations of Williams et al., whereas it is 12.5 m in the reality when measured from trunks of the eastern trees (Fig. 1b and c).

After eliminating the shielding effect from the time series using the correction factors calculated by Williams et al. (2005), a slight but statistically significant decrease still remains in the time series of annual mean PG values (Table 3 and Fig. 5a). However, if the correction factors calculated in the present study are applied, the long-term reduction vanishes entirely (Table

**Table 3.** Changes of PG at NCK as estimated by linear approximation. Percental change gives the ratio of the PG value estimated in the last year of the considered time period to the value obtained in the first year of the considered time period using linear approximation. Percental change rate is the percental change divided by the number of years in the considered time period. The uncertainties are deduced (propagated) from the uncertainties of the corresponding tree heights and linear fit and carry the uncertainties of the model.

| | Percental change | Percental change rate | $r^2$ | p-value |
|---|---|---|---|---|
| Uncorrected annual means (1962–2009) | -58 % | -1.2 % year$^{-1}$ | 0.96 | p<0.001 |
| Corrected annual means (Williams et al., 1962–2002) | -12 % | -0.3 % year$^{-1}$ | 0.41 | p<0.001 |
| Corrected summer means (present study, 1962–1985) | +10±4 % | +0.4±0.2 % year$^{-1}$ | 0.14 | p>0.05 |
| Corrected annual means (present study, 1962–1985) | +21±2 % | +0.9±0.1 % year$^{-1}$ | 0.49 | p<0.001 |
| Corrected winter means (present study, 1962–1985) | +22±5 % | +0.9±0.2 % year$^{-1}$ | 0.32 | p<0.05 |
| Corrected summer means (present study, 1986–1997) | +42±3 % | +3.5±0.2 % year$^{-1}$ | 0.57 | p<0.05 |
| Corrected annual means (present study, 1986–1997) | +9±2 % | +0.8±0.1 % year$^{-1}$ | 0.33 | p>0.05 |
| Corrected winter means (present study, 1986–1997) | -1±3 % | -0.1±0.3 % year$^{-1}$ | 0.003 | p>0.05 |
| Corrected summer means (present study, 1997–2009) | -23±3 % | -1.8±0.3 % year$^{-1}$ | 0.81 | p<0.001 |
| Corrected annual means (present study, 1997–2009) | -19±2 % | -1.5±0.2 % year$^{-1}$ | 0.9 | p<0.001 |
| Corrected winter means (present study, 1997–2009) | -19±4 % | -1.5±0.3 % year$^{-1}$ | 0.66 | p<0.001 |

3 and Fig. 5a). Instead of a reduction, the corrected annual averages exhibit a slight increase between 1962 and 1985 (Table 3 and Fig. 5b). The raising trend seems to be somewhat damped after 1985 (at least in case of the annual and winter means) but the enhancement tends to continue until 1997 (Table 3). After 1997, a fairly steep reduction starts in the PG at NCK (Table 3).

The reconstructed long-term variation of PG at NCK is significantly different from previous results and variations observed at some other stations, e.g., Eskdalemuir (U.K.), Serra do Pilar (Portugal) (Märcz and Harrison, 2003, 2005). At the Świder Geophysical Observatory, Poland (52°07′ N, 21°14′ E), the PG has been measured since 1958 (Kubicki, 2000; Dziembowska, 2009; Kubicki et al., 2021), which allows us to compare our results to independent measurements made in the same region, i.e.,

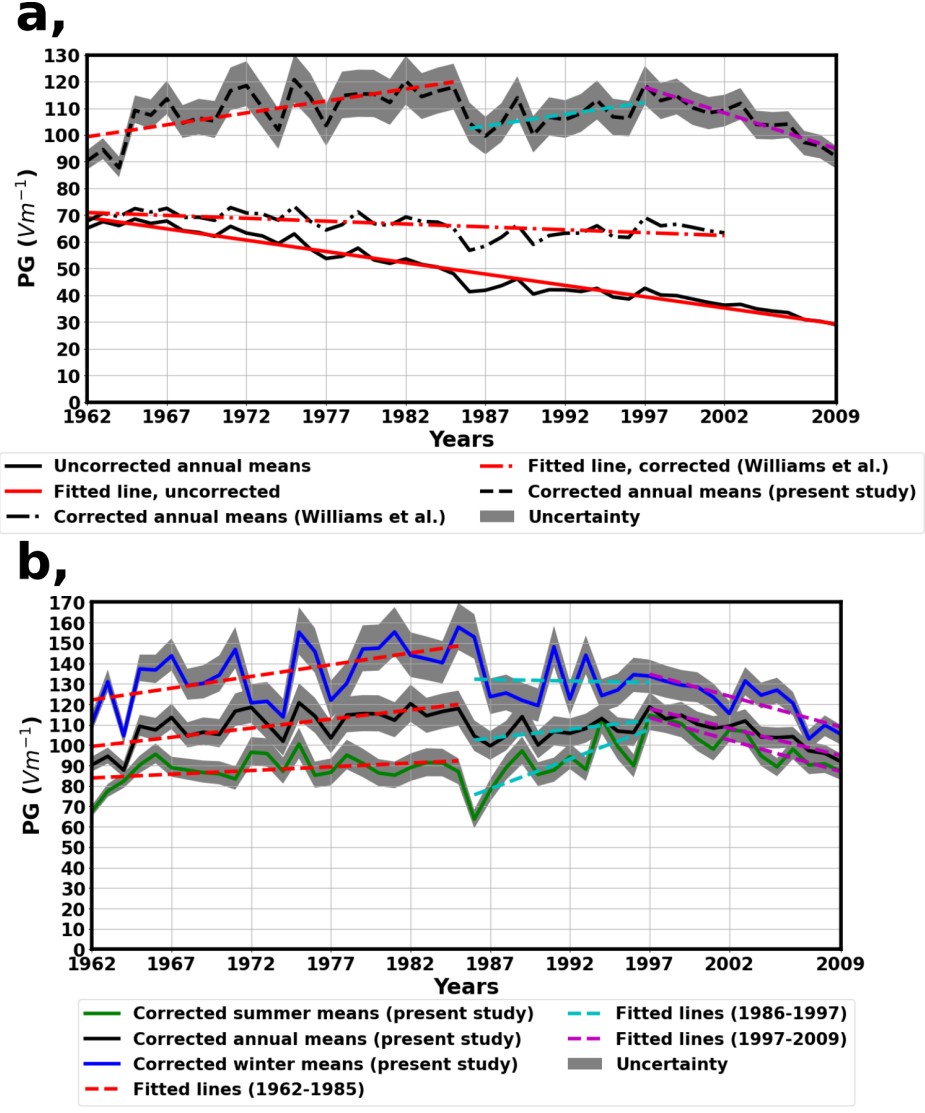

**Figure 5.** PG values recorded at NCK. **(a)** Annual means of uncorrected and corrected records. **(b)** Annual, summertime, and wintertime averages of the corrected records. Dashed lines correspond to linear regressions on the time series (see Table 3).

Central Europe. There are similar increasing (1962–1985 and 1987–1997) and decreasing (2003–2007) trends in the Świder data which appear in the NCK PG time series only after the removal of the shielding effect (Fig. 5 and Fig. 6).

Regarding the last period, a similar decreasing trend in the PG has been reported based on the data of an array of field mills at the Kennedy Space Center (KSC, Florida, USA) (Lucas et al., 2017). The magnitude of the decrease between 2003 and 2009 is very much alike at all three (NCK, KSC, and Świder) locations (see Fig. 7 in (Lucas et al., 2017)). The fact that this decreasing trend since 2003 can be observed in three PG datasets recorded at remote sites affirms the global origin of the reduction and

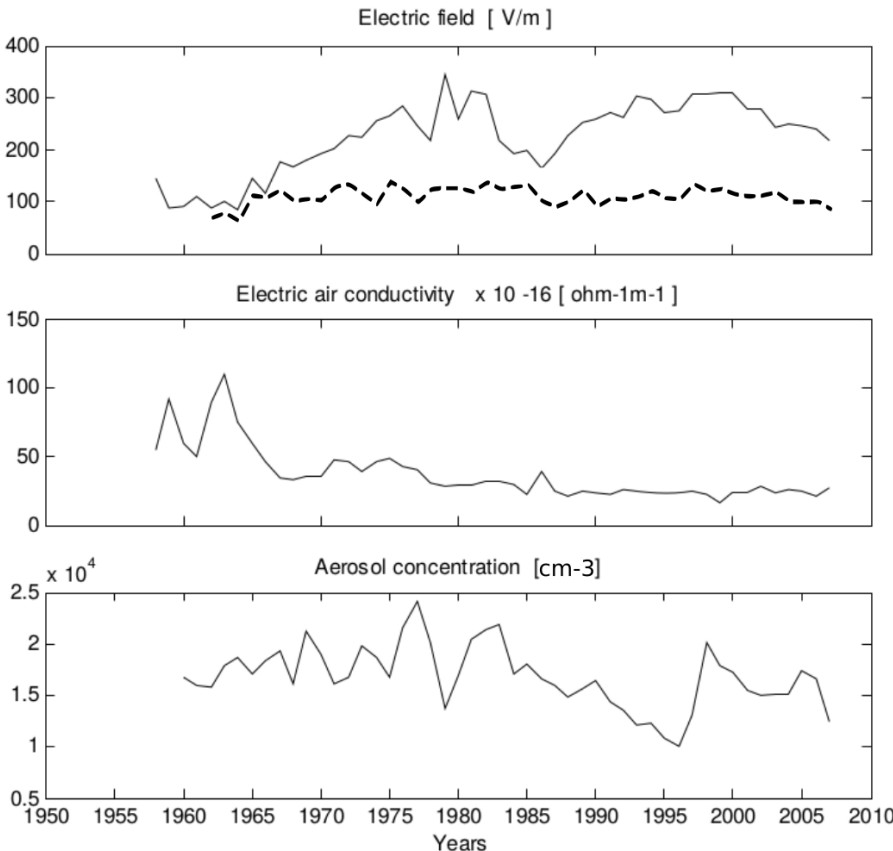

**Figure 6.** Long-term variations of the electric field, electric conductivity of air, and aerosol concentration at Świder Observatory (Poland). Dashed line on the uppermost panel denotes the corrected annual means measured at NCK. Adopted with permission from Fig. 4. in (Dziembowska, 2009).

assures us that the model described in this study can successfully remove the shielding effect of local trees from the data thus revealing potentially global signals.

The observed similarities indicate that the modelling and removal of the shielding effect used in the present study is a valid approach. The observed similarity of long timescale PG variations at the two separated observatories (Świder and NCK) demonstrates that these time series show variations of the PG on large spatial scales. This is further supported by the observation that yearly averaged local variation of air conductivity and aerosol concentration in Świder does not seem to appear unambiguously in the corresponding PG measurements (Fig. 6). Air conductivity and PG are usually negatively correlated

while the aerosol concentration and PG are positively correlated in most cases but there is not such a clear relationship in the long-term data in Świder (Fig. 6) (Kubicki et al., 2016, 2021). Note that the correspondence between these parameters in Świder could have been evaluated more completely if the local distribution of charged aerosol sizes in Świder were known as the conductivity of the air also depends very much on that (Dhanorkar and Kamra, 1997). The anti-correlation of the measured

PG and air conductivity values in Świder can be observed clearly on shorter timescales of few hours as it was shown in the case of the Chernobyl accident (Warzecha, 1987).

Note that the absolute PG values of the two datasets differ significantly. Much higher values were observed in case of the measurements at Świder, although at the initiation of the measurements at NCK (in 1962) the PG at NCK and at Świder had almost the same magnitude (Fig. 5 and Fig. 6). Such difference is characteristic for measurements of different sensitivities. However, both time series are based on carefully calibrated records. A reason behind this difference in magnitude might be that the Geophysical Observatory of Świder is located near (ca. 15 km away) Warsaw, the capital of Poland and is surrounded by settlements. The anthropogenic pollution is likely higher in Świder than in NCK as the latter is embedded in a Natural Park. The nearest small village is in a distance of 1.3 km and the nearest (small) town (Sopron) is 10 km away from NCK. Higher pollution decreases the air conductivity and so it causes higher PG values. Air conductivity in Świder after the perturbed period of atmospheric nuclear weapon tests (1958-1965) is around $3 - 4 \times 10^{-15} \mathrm{~Sm}^{-1}$ (Fig. 6) (Kubicki et al., 2021) whereas the average fair weather air conductivity is greater by one order of magnitude (around $1.3 \times 10^{-14} \mathrm{~Sm}^{-1}$) (Rycroft et al., 2000). Please note that PG measured at different sites can have highly different magnitudes. For instance, in a paper where 17 PG stations were compared, the non-disturbed PG median of all the investigated data ranged from $21 \mathrm{~V\,m}^{-1}$ to $404 \mathrm{~V\,m}^{-1}$ (Nicoll et al., 2019). The high variability of PG at different sites, alongside with the different sensitivity of instruments at NCK and Świder, are likely to be the reason behind the different absolute PG values at the two sites.

## 4.2 Seasonal variation of the corrected potential gradient

There is a global minimum in the summer means time series in 1986 which appears in the annual means as well (Fig. 5b). These anomalously low values were caused by the Chernobyl power plant accident in April, 1986. As radionuclides were transported by winds from Chernobyl to NCK, deposited radioactive fallout ionized the air near the surface and caused a reduction in the PG. Similar phenomenon was observed at several other atmospheric electricity monitoring sites across Europe (Israelsson and Knudsen, 1986; Warzecha, 1987; Retails and Pitta, 1989; Tuomi, 1989). Note that no such reduction can be seen in wintertime PG data (Fig. 5b). This difference may be caused by the snow which could cover radionuclides on the ground and prohibit ionizing radiation from reaching the air (Tuomi, 1982). Additionally, since the accident occurred in April and winter means were taken from the values in December, January, and February this difference may also indicate the recovery of the environment after the contamination.

The increase in the PG before 1985-86 is stronger in wintertime data than in summertime data (Table 3, Fig. 5b). On the other hand, the growing trend at NCK seems to cease in the winter averages after 1986. From 1986, the summer and annual PG datasets show a moderate increase until 1997. This increase is more pronounced in case of the summertime values where the change rate is practically eight times larger compared to the previous time period (Table 3). This means that the growing trend in the annual averages in this period is practically driven by the changes in the summertime values. The decrease, which can be observed unambiguously after 1997, appears in all considered time series (Fig. 5b, Fig. 6). The change rate of the trend is larger and more robust in summertime data than in the wintertime time series (Table 3). The origin of these variations is not clear and finding it is out of the scope of this paper. Nevertheless, the characteristically different variation of summertime and

wintertime PG averages emphasizes the importance of considering the seasonal means in the interpretation of long-term PG variations separately.

If the whole investigated time period (1962–2009) is considered, the month of the largest PG mean in the year tends to be in winter (December, January, February), while the month of the smallest PG is mostly in late spring and in the summer (May, June, July) (Fig. 7a and b). This is frequently found at continental sites. The higher concentration of aerosols due to the greater pollution in winter results in reduced air conductivity and enhanced PG compared to summertime averages (Chalmers, 1967, pp. 168–169).

Note that the characteristic period of smallest PG values of the year is shifted to late spring and early summer, i.e., May and June in most of years at NCK (Fig. 7a). On the other hand, minimum was also found in autumn or in winter in some years. This behaviour has been already reported on the basis of a shorter time period (1962–2001) (Märcz and Harrison, 2003). The origin of this peculiarity remains to be explained yet, but most probably local climate characteristics contribute to it. As it was pointed out by Märcz and Harrison (2003), summer PG values reflect more the local and regional changes, whereas winter

values are generally more sensitive to global variations. Figure 7b demonstrates that long term variations of the PG averaged over the summer and winter months show practically the same trends as the variation of the yearly minimum and maximum of monthly averages, respectively. Nevertheless, Fig. 7a calls the attention on that joint evaluation of these two parameters is required in order for the results to be interpreted correctly.

## 5   Summary and conclusions

Two PG measurement campaigns were carried out at NCK in order to assess the shielding effect of nearby trees. One in the summer (03–04. August 2016) and one in wintertime (24–25. January 2017). It was found that the shielding effect due to nearby trees reduces the PG significantly, even causing near-zero values at some points. Approaching the trees, the shielding effect becomes more and more pronounced (Fig. 4a, b, and c).

     A numerical model was built to quantify the electrostatic shielding effect of the trees. The geometry was set up in such

a fashion that it mimics the real configuration of the measurement site as much as it is possible (Fig. 1b and c). On-site measurements showed that the model is generally in good accordance with the observed PG values and describes the shielding effect adequately (Fig. 4). Input parameters of the model were determined from age-height curves for tree species that are present at NCK based on averaged measurements in Hungary (Fig. 3).

     Based on in situ measurements and numerical modelling, the shielding effect of the trees was quantitatively determined

(Table 1) and the PG values measured at NCK in fair weather conditions were corrected for this bias. It can be concluded that the variation of the corrected PG time series does not support the presence of a long-term decreasing trend in the examined time range (from 1962 to 2009; Fig. 5 and Table 3). On the contrary, the PG time series measured at NCK exhibit an increase between 1962 and 1985, a moderate increase or a stagnating trend in the 1986–1997 time interval, and a decreasing trend after 1997 (Fig. 5 and Table 3).

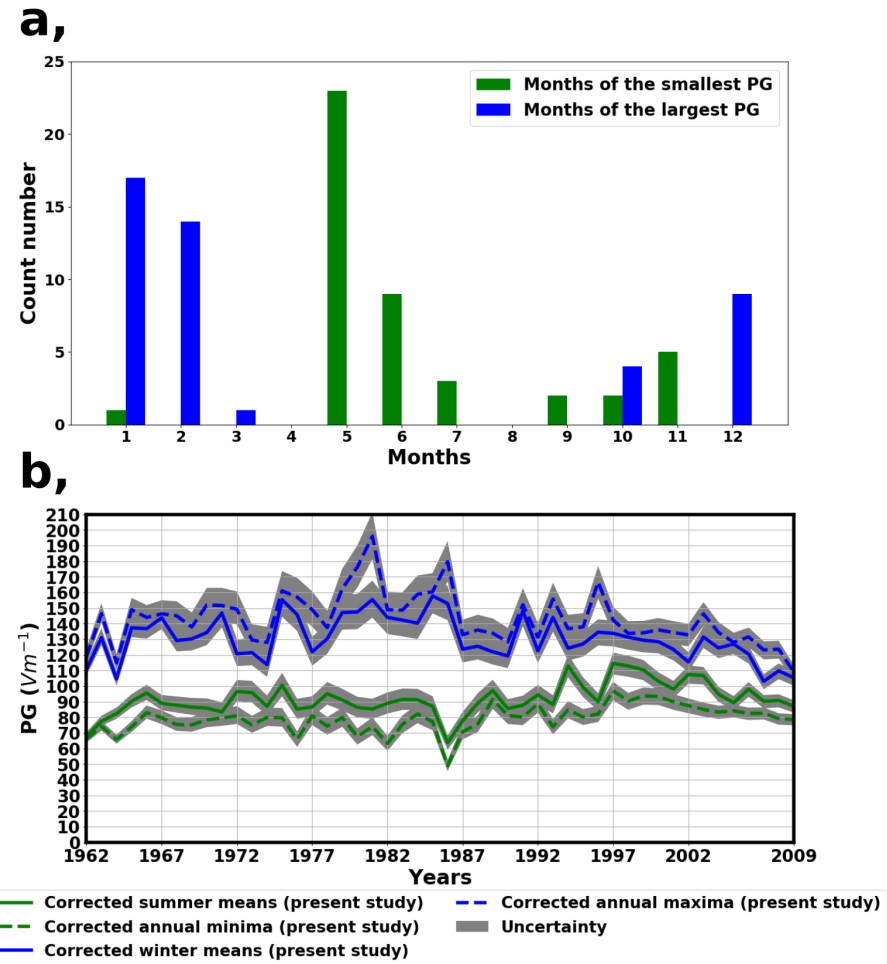

**Figure 7. (a)** The number of months with the annual greatest or smallest PG monthly mean value between 1962 and 2009. **(b)** Comparison of the annual minima/maxima and summer/winter means PG time series.

This behaviour is in good accordance with the long-term PG measurements from Świder, Poland (Dziembowska, 2009) in the same region, i.e., Central Europe. The similarity of the Świder PG time series affirms that the numerical model presented in this study reliably quantifies the time-dependent local shielding effect and the corrected annual average PG time series mirror the real long-term PG variations at NCK.

The PG reduction that is revealed in the corrected NCK data series after 1997 is present both in the Świder measurements (after 2000) and in the PG data recorded by an array of field mills at the Kennedy Space Center in Florida, USA (Lucas et al., 2017). Coherent observations in three well separated stations suggest that this is a globally present trend.

The results basically confirm the conclusion of the paper by Williams et al. (2005), i.e., that the long-term reduction in the NCK PG time series was caused by the time-dependent shielding effect of nearby trees. However, characteristics of the

investigated measurement site have been taken into account more rigorously in the present study, so the results presented here are expected to be more accurate.

The corrected PG data measured at NCK show the expected seasonal variation with more enhanced values in winter than in summertime (Fig. 5b). However, the month of the greatest mean in the year tends to occur late spring–early summer (mostly in May and June) at NCK rather than only in the summer months June, July, and August (Chalmers, 1967, pp. 168–169) (Fig. 7a). We suspect that local and regional effects cause this behaviour at NCK but further studies are needed for clarification.

It was demonstrated that the long-term variation of summer and winter PG averages is very much different at NCK (Fig. 5b and Fig. 7b). The variation of annual PG means can be driven by either the summertime or the wintertime trends whichever is dominant in a given time period (Table 3).

One important conclusion of this work is that the electrostatic shielding effect of nearby objects may have a large influence on PG measurements and can entirely suppress long-term trends (i.e., over 5–10 years) in global signals. It is suggested to perform parallel PG measurements similar to those described in this study to quantify any shielding effect. According to analytic calculations, thin conducting objects (such as a metallic pole or a fence) can distort the ambient atmospheric electric field up to 5 % in a distance of 3 times their height and up to 1 % in a distance of 10 times their height (Lees, 1915). If the object is wider (like a forest or a bigger building) the distortion is around 5 % in a distance of 5 times the height of the object and 1 % in a distance of 33 times the height of the object (Benndorf, 1900; Lees, 1915). These calculations can help to place PG instruments in any site appropriately. Validation of the measurements can be done using PG data measured independently at displaced stations and modelling the geometry of the site. In any case, the environmental conditions of the measurements must be considered and taken into account at the evaluation of the results.

*Author contributions.* AB carried out most of the analysis and created the paper together with JB's active help. JB and VB assisted with the on-site measurements and their invaluable consultancy proved to be a tremendous help in the preparation of the manuscript. TH provided the tree height curves which were crucial in finishing successfully the study. All co-authors helped in the interpretation of the results, read the paper and commented on it.

*Competing interests.* The authors declare that they have no conflict of interest.

*Acknowledgements.* The authors wish to express their sincere gratitude to Tamás Bozóki who shared his valuable comments and suggestions regarding the manuscript. Scientific discussion connected to the topic of this paper was supported by the COST Action CA15211, „Atmospheric Electricity Network: coupling with the Earth System, climate and biological systems" (ELECTRONET), CA15211. The work of József Bór was supported by the National Research, Development and Innovation Office, Hungary-NKFIH, K115836.

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
