# Peer review of "Revisiting the long-term decreasing trend of atmospheric electric potential gradient measured at Nagycenk, Hungary, Central Europe"

_Annales Geophysicae, 2021_

## Author Response (AR1)

**Changes in the revised manuscript**

The initially submitted manuscript has been thoroughly re-checked and revised according to the reviews. First of all, we would like to express our gratitudes towards the Referees and the Editor as they helped us to change the manuscript for the better. Several minor changes were made, typos were corrected, references were added, some of the affiliations of the authors were changed, numerical values were updated. The following are the major changes made in the manuscript according to the reviews:

- A new method of fair weather data selection was used according to our response to the first Referee. This new method takes into consideration the year-to-year changes in the PG (atmospheric electric potential gradient) data. All the calculations and numerical values were updated according to this new fair weather selection criterion. This issue is described in Section 2.1.

- More attention was payed to the different uncertainties and error propagation throughout the entire manuscript.  The uncertainties were taken into account in cases of Fig. 2 and 4 and Tables 1 and 3. In case of Table 3, the usage of p-value was corrected.

- A new method of multivariate linear regression was used in case of Fig. 2. Owing to this new method, the deduced linear regression parameters of the data measured on $4^{th}$ August 2016 were retained and data from $4^{th}$ August were used in the model validation (Fig. 4b). Please note that in the Authors's response (response to major point one of the $1^{st}$ Referee), initially we said that we could not retain some of the PG values recorded on $4^{th}$ August. With this new multivariate linear regression method however, we were able to retain those data and include them in the model validation (please check the explanation in Sections 2.3 and 3.1).

- The question of conductivity and dielectric constants was revised according to the report of the second Referee. The inappropriate usage of the so called "perfect conductors" was omitted. The model calculations were done with dielectric constant values derived from the relevant literature. Then the need of the adjustment of these dielectric constants and the usage of the so called effective (fitted) dielectric constants were explained in the text. Please check Sections 2.3 and 3.1.

- More attention was payed to the question of aerosol concentration in Swider as well. However, please note that this subject lies out of the scope of this study so this problem was elucidated briefly. Please refer to Section 4.1 where the difference between the magnitude of PG values recorded at NCK and Swider is explained as well.

- The last point of the conclusions in Section 5 was specified and extended. A recommendation about locating PG instruments so as to avoid any unwanted shielding effect was made.

- The uncertainty of the tree heights used as an input of the numerical model was explained in Section 2.4.

- Four relevant references were added (Boggs and Rogers, 1990; Dhanorkar and Kamra, 1997; Kubicki et al, 2016, 2021).

Othe minor points were revised according to the Author's response to the referee reports.

Explanations for the marked-up version of the revised manuscript: Green colored text means deleted sections while red means new, revised text. The following figures and tables were changed during the revision: Figures 2, 4, 5, 6, 7; Tables 1, 3.